# Thyroid Pathology in End-Stage Renal Disease Patients on Hemodialysis

**DOI:** 10.3390/diagnostics10040245

**Published:** 2020-04-23

**Authors:** Laura Cotoi, Florin Borcan, Ioan Sporea, Daniela Amzar, Oana Schiller, Adalbert Schiller, Cristina A. Dehelean, Gheorghe Nicusor Pop, Andreea Borlea, Dana Stoian

**Affiliations:** 1PhD School Department, “Victor Babes” University of Medicine and Pharmacy Timisoara, 2nd Eftimie Murgu Square, 300041 Timisoara, Romania; cotoi.laura@umft.ro; 2Analytical Chem. and Toxicology Department, “Victor Babes” University of Medicine and Pharmacy Timisoara, 2nd Eftimie Murgu Square, 300041 Timisoara, Romania; cadehelean@umft.ro; 3Internal Medicine 2nd Department, “Victor Babes” University of Medicine and Pharmacy Timisoara, 2nd Eftimie Murgu Square, 300041 Timisoara, Romania; isporea@umft.ro (I.S.); adalbert.schiller@yahoo.com (A.S.); 4Endocrinology Department, “Victor Babes” University of Medicine and Pharmacy Timisoara, 2nd Eftimie Murgu Square, 300041 Timisoara, Romania; amzar.danielageorgiana@gmail.com (D.A.); Stoian.dana@umft.ro (D.S.); 5Dialysis Medical Center B Braun Avitum, 636, 307350 Remetea Mare, Romania; oana.schiller@bbraun.com; 6Centre for Modeling Biological Systems and Data Analysis Department of Functional Sciences, “Victor Babes” University of Medicine and Pharmacy Timisoara, 2nd Eftimie Murgu Square, 300041 Timisoara, Romania; pop.nicusor@umft.ro

**Keywords:** nodular goiter, hemodialysis, thyroid disease, end-stage renal disease

## Abstract

Objectives: Chronic kidney disease is a rising cause of morbidity and mortality in developed countries, including end-stage renal disease (ESRD). The prevalence of thyroid comorbidities in persons with chronic kidney disease is documented higher than in normal population. The study aims to investigate the prevalence of morphological and functional thyroid disorders in patients with chronic kidney disease, with renal replacement therapy (hemodialysis). Methods: A cross-sectional study was performed on 123 consecutive patients with chronic kidney disease stage 5, on hemodialysis during a period of one month (May 2019–June 2020). All patients were enrolled for maintenance hemodialysis in B Braun Hemodialysis Center Timisoara and were examined on conventional 2B ultrasound. Thyroid blood tests were done, including serum free thyroxin (FT4), free triiodothyronine (FT3) and thyroid-stimulating hormone (TSH) at the time of starting hemodialysis. Results: We evaluated 123 patients (male to female ratio 70/53) mean age 62.2 ± 11.01, mostly above 65 years old, enrolled in the end-stage renal disease program, on renal replacement therapy. From the cohort, 76/123 presented thyroid disease, including autoimmune hypothyroidism, nodular goiter or thyroid cancer. Among them, 63 patients presented nodular goiter, including 3 thyroid cancers, confirmed by surgery and histopathological result, 22 patients had thyroid autoimmune disease. The serum thyroid-stimulating hormone levels found in the cohort was 3.36 ± 2.313 mUI/mL, which was in the normal laboratory reference range. The thyroid volume was 13 ± 7.18 mL. A single patient in the cohort presented Graves Basedow disease, under treatment and three patients present subclinical hyperthyroidism. We have found that thyroid disease risk is increased by 3.4-fold for the female gender and also the increase of body mass index (BMI) with one unit raises the risk of developing thyroid disease with 1.083 times (*p* = 0.018). Conclusion: To conclude, this study aimed to quantify the prevalence of thyroid disease in end-stage kidney disease population, especially nodular goiter, important for differential diagnosis in cases with secondary hyperparathyroidism. Thyroid autoimmune disease can be prevalent among these patients, as symptoms can overlap those of chronic disease and decrease the quality of life. We have found that thyroid disease has a high prevalence among patients with end-stage renal disease on hemodialysis. Thyroid goiter and nodules in ESRD patients were more prevalent than in the general population. Clinical surveillance and routine screening for thyroid disorders can improve the quality of life in these patients.

## 1. Introduction

Endocrine disorders are highly common endocrine complications among patients with chronic kidney disease, including those receiving dialysis [1]. 

Chronic kidney disease (CKD) represents a worldwide public health problem and it is defined by the National Kidney Foundation (NKF) Kidney Disease Outcome Quality Initiative (K/DOQI) as kidney damage for more than three months with structural or functional abnormalities, with or without decreased glomerular filtration rate (GFR), manifested by pathological abnormalities or markers of kidney damage, or by GFR < 60 mL/min/1.73 m^2^. CKD is classified into five stages upon GFR value [2]. Kidney Disease Improving Global Outcomes (KDIGO) guidelines characterize CKD by using markers of kidney damage (proteinuria and glomerular filtration rate). Chronic kidney disease is defined by the presence of both glomerular filtration rate less than 60 mL/min and albumin greater than 30 mg per gram of creatinine, along with abnormalities of kidney structure or function for more than three months [3]. End-stage renal disease (ESRD) is defined as a GFR less than 15 mL/min or CKD stage 5 [3,4].

There is a wide array of methods to support the renal function, by delivering ongoing organ support. Renal replacement therapy (RRT) is defined by using extracorporeal methods (hemodialysis) or paracorporeal methods (peritoneal dialysis) [2,5].

Chronic kidney disease exhibits multiple endocrine and metabolic effects. One of the most common is on bone metabolism, causing chronic kidney disease mineral bone disorder (CKD-MBD) and secondary hyperparathyroidism [6]. Another important endocrine pathology present in patients with chronic kidney disease is hypothyroidism and nodular goiter. There are important studies in the literature that support this fact [7,8,9,10,11]. The presence of nodular goiter in patients with chronic kidney disease brings an important burden on the clinician when facing a patient with secondary hyperparathyroidism. Evaluation of the percentage of thyroid disease, especially nodular thyroid goiter among patients with CKD could contribute to the correct diagnosis and treatment of these patients.

The interaction between the thyroid and the kidney in each other’s functions is known for many years [8]. Thyroid hormones (TH) play an important role in kidney physiology. Their actions on growth, development and maintenance of kidney homeostasis are known and have been discussed in literature studies [12]. Thyroid dysfunctions, both hypothyroidism, and hyperthyroidism have an important effect on kidney function, as well as cardiovascular alterations [12].

Chronic kidney disease causes disturbances in the hypothalamus-pituitary–thyroid axis and TH peripheral metabolism [8,13,14,15]. Studies have shown that uremia plays an important role in patients with ESRD, having a direct relationship with the size of thyroid and the prevalence of thyroid goiter [7,8,16]. In addition, thyroid nodules and thyroid cancer have a higher prevalence in uremic patients, than in the general population, especially in women [12,17].

Peripheral thyroid hormones (T3 and T4) can present low concentrations in serum, because of the hormonal removal during dialysis, decreased T3-binding capacity, altered hormonal catabolism, increased iodine store in thyroid gland, serum thyroid autoantibodies, and decreased peripheral conversion [7,8,12,18]. Low T3 syndrome has been observed in patients with ESRD [7,8,15,19], caused on one hand by the diminished peripheral conversion of T3 from T4 and on the other hand by the chronic metabolic acidosis [20,21]. 

End-stage renal disease can overlap the symptoms of hypothyroidism, such as fatigue, lethargy and cognitive dysfunction, as the prevalence data previously known are on relatively small cohorts and there are little data about the severity of thyroid abnormalities in these patients. A relatively high prevalence of thyroid goiter and an increase in thyroid gland have been reported [22]. An increased prevalence of hypothyroidism and subclinical hypothyroidism has been suggested in patients with ESRD [7,22]. Thyroid hormone dysfunction has been reported in ESRD patients, including low peripheral thyroid hormones and thyroid stimulating hormone [7,23].

The general population prevalence of overt hypothyroidism accounts for 0.2% to 5.3% in Europe [24,25,26] and 0.3% to 3.7% [27,28] of the general population. In the CKD population, there is a high prevalence of primary hypothyroidism, as it increases with the decline of glomerular filtration rate (GFR) [29], especially in the subclinical form of hypothyroidism. Literature studies have shown a prevalence of 7% of subclinical hypothyroidism in patients with estimated GFR < 90 mL/min/1.73 m^2^, and the prevalence increased to 17.9% in subjects with GFR < 60 mL/min per 1.73 m^2^ [30,31,32]. 

The prevalence of hypothyroidism among ESRD patients on hemodialysis (HD) in previous studies was variable. One study reported a prevalence of subclinical hypothyroidism in HD patients of 21.8% compared with 7.14% in the control group [30], another study reported a prevalence of hypothyroidism in ESRD patients of 2.6% compared to 1.1% in the control group [7]. A study conducted on Iranian patients reported a prevalence of nodular goiter of 27.9% in HD patients versus 3.5% in the control group and hypothyroidism in 18.6% in HD patients versus 8.1% in the control group [33]. 

The prevalence of hyperthyroidism among patients with CKD is similar to the prevalence in the general population, around 1% [29,34,35,36].

Thyroid cancer prevalence was reported higher among ESRD patients on hemodialysis, with a positive correlation between the occurrence of thyroid cancer and duration of dialysis, and a higher rate of multi-focality and also in post-transplant patients [36,37,38,39].

This study was designed to evaluate the prevalence of thyroid morphological changes and hormonal dysfunction in hemodialysis patients and to investigate if there are any associations between dialysis duration, age, body mass index or biochemical serum changes and the prevalence of thyroid disease.

## 2. Material and Methods

A cross-sectional observational single-center study was carried out in B Braun Dialysis Center Timisoara, from May 2019 to June 2019. One hundred and twenty-three subjects were enrolled in this study. All patients were on chronic hemodialysis treatment with different primary causes of renal failure, with a mean duration of hemodialysis therapy 5.6 ± 4.89 years. All patients were clinically stable and free of active infections. Hemodialysis was performed three times a week, four hours per session.

The cohort consisted of adult patients with confirmed end-stage renal disease, on hemodialysis with previous or no history of thyroid disease. We excluded patients with previous thyroid surgery, with total thyroidectomy. The study was approved by the Ethics Committee, of the Dialysis Center (IIT2/24.04.2019), and all patients signed written informed consent. The study was following the Ethics Code of the World Medical Association (Declaration of Helsinki, Seoul, Korea, October 2008). 

A complete clinical and biochemical profile was obtained for each patient. Blood tests were performed in all patients before the onset of the dialysis session and heparin administration. Serum values of blood urea nitrogen, creatinine, glucose, total cholesterol, triglyceride, total calcium, and phosphorus were examined by an auto analyzer (Dimension RxL Max Integrated Chemistry System, Siemens, mass spectrophotometry method). 

The adequacy of hemodialysis was calculated as a fractional clearance index for urea (Kt/V) and urea reduction ratio by using a single compartment dialysis urea kinetic model.

Following measurements were taken in consideration: calcium (reference range 8.5–10.2 mg/dL) —mass spectrophotometry method, PTH (reference range 15–65 pg/mL)—immunochemistry with enzyme chemiluminescence immunoassay (ECLIA); vitamin D (reference range 30–100 ng/mL)—immunochemistry with ECLIA, free triiodothyronine (FT3 reference range 3.4–6.8 pmol/L) and free thyroxin T4 (FT4 reference range 12 to 22 pmol/L—radioimmunoassay (RIA), thyroid-stimulation hormone (thyroid-stimulating hormone (TSH) reference range 0.27–4.20 µU/mL) was measured by chemiluminescence immunometric assay (IRMA), and the index of urea Kt/V directly calculated for each patient (Table 1). 

Conventional B-mode thyroid ultrasound was performed in all cases on Aixplorer Mach 30 (SuperSonic Imagine, Aix-en-Provence, France) using a high-resolution linear transducer of 18-5 MHz, with the patient resting in supine position with regular breathing, applying an adequate amount of ultrasound gel.

Using grey scale US, we evaluated the following parameters: thyroid dimensions (two dimensions in transverse scan and one dimension in longitudinal), thyroid volume measured by the sum of lobes volumes, thyroid parenchyma echogenicity, vascularity and the presence of nodules. The volume for each lobe was calculated by the ellipsoid formula; V = a × b × c × n/6 (n—correction factor measuring 0.529 [40]). All patients were clinically evaluated and with ultrasonography by two practitioners, one with over 15-year experience in thyroid, parathyroid and neck ultrasound.

The normal range of serum thyrotropin was considered according to laboratory results as 0.27 to 4.2 milli-international units per liter (mUI/L), and free triiodothyronine index was considered from 3.4 to 6.8 pmol/L. We defined thyroid disorders into four categories according to biochemical serum concentrations of thyroid hormones. Euthyroidism was considered if serum concentrations of TSH, FT3, FT4 were normal, hypothyroidism was considered in an elevation in TSH > 10 mIU/L with reduced FT3 or FT4 levels. Subclinical hypothyroidism was in in TSH > 10 mIU/L with normal FT3 or FT4 levels. Hyperthyroidism was in TSH < 0.01 mIU/L with high concentrations of FT3 or FT4. Subclinical hyperthyroidism was in TSH < 0.01 mIU/L with normal concentrations of FT3 or FT4. Euthyroid sick syndrome was defined as normal TSH and FT4 levels and T3 level < 2.7 pmol/L (Table 2).

According to the ultrasound evaluation and laboratory results, nodular goiter was considered in the presence of one node in any of the thyroid lobes. 

Hashimoto thyroiditis was characterized by a heterogeneous hypoechoic gland with echogenic septa and increased vascularity associated with elevated thyroid antibodies (TG Ab, TPO Ab). We have found 63 patients with nodular goiter (51.2%) and 22 patients with autoimmune thyroid disease (17.8%). Twelve patients had both nodular goiter and autoimmune thyroiditis. Three patients underwent total thyroidectomy and presented thyroid cancer—papillary thyroid carcinoma at pathology report. Thirteen patients presented previous parathyroid surgery for secondary hypothyroidism, but with intact thyroid gland (Table 3).

### Statistical Analysis 

Continuous variables were presented as mean and standard deviation (SD) and categorical variables were presented as frequency and percentages. We performed descriptive and inferential statistical analysis to summarize the characteristics of the study population. To compare patients with or without autoimmune thyroiditis and nodular goiter specific group-pairs with each other, we used the t test. For the nominal variable we employed cross-tabulation with Chi-square test and the odds ratio was calculated. To highlight the relationship between numerical variable, we performed Pearson’s correlation. To find the independent predictive factors for thyroid disease we employed logistic regression. Akaike information criteria (AIC) were used in order to determine the best model and the quality of the model was described using Nagelkerke’s R2 for the accuracy of prediction. The predictors, in the final regression equations, were accepted according to a repeated backward-stepwise algorithm (inclusion criteria *p* < 0.05, exclusion criteria *p* > 0.10) to obtain the most appropriate theoretical model to fit the collected data. Data were collected and analyzed using SPSS v.25 (Statistical Package for the Social Sciences, Chicago, IL, USA). A *p* value of < 0.05 was considered to indicate a statistically significant difference.

## 3. Results 

We evaluated 123 patients with end stage renal disease, with renal replacement therapy, with hemodialysis three times a week, mean age 62.2 ± 11.01 years. The mean duration of hemodialysis therapy was 5.6 ± 4.89 years.

After dividing the patients according to laboratory results we have found that 74.5% of patients from our study group were euthyroid, 24.4% percent were hypothyroid (16 females and 14 males), and 4 patients (3.3%) had subclinical hyperthyroidism, but only one patient was previously known with Graves Basedow disease, the other three patients presented subclinical hyperthyroidism. A total of 57 (46.34%) patients presented concentrations of FT3 lower than 2.7 pmol/L, but only 40 out of 57 presented thyroid disease.

According to ultrasound results, 48.7% of patients have a normal thyroid appearance, 51.2% have a nodular goiter, defined as the presence of minimum one node in any thyroid lobe and 17.8% had autoimmune thyroiditis, with positive antibodies. 

On ultrasound evaluation however, we have found 41 patients with diffusely hypoechoic gland or multiple hypoechoic foci in the thyroid parenchyma and normal thyroid antibodies concentrations.

In the nodular goiter group, 7 patients underwent total thyroidectomy and 3 patients presented papillary thyroid carcinoma on pathology report after surgery. A total of 31 patients had one nodule (49.2%), 16 patients had two nodules present on ultrasound (25.3%) and 16 patients, including those who underwent surgery, had minimum three nodules on ultrasound evaluation (25.3%).

The mean diameter of the nodules was 4.7 ± 11.1 mm, the maximum size found on ultrasound was 24.6 mm and the minimum dimension was 2 mm. The echogenicity on most nodules was hypoechoic, a part with cystic appearance, a part homogeneously solid and hypoechoic appearance and some had a mixed appearance.

Among the patients with nodular goiter, 34 were females (53.9%) and 29 were males (46.03%), and in the autoimmune group, 12 were females (54.4%) and 10 were males (45.4%). From the group that had thyroid ultrasound appearance for thyroid disease, but negative antibodies, 33 were women (52.3%) and 30 were men (47.61%). 

Secondary hyperparathyroidism was found in 59 patients, with a total of 97 hyperplastic parathyroid glands were visible on ultrasound [41]. In this group, 30 patients also presented nodular goiter (Figure 1) and 13 patients presented autoimmune thyroiditis. Thyroid nodules frequently pose a dilemma to the clinician, firstly in respect to the nature of the nodule and also when faced with patients with parathyroid hyperplasia. The prevalence of thyroid disease, especially nodular thyroid goiter among patients with ESRD could contribute to the correct diagnosis and treatment of these patients.

Considering TSH serum concentrations and Kt/v ratio from the hemodialysis session, we have found no significant correlation (*r* = 0.005; *p* = 0.957). We found a significant correlation between TSH concentration and age (*r* = 0.185, *p* = 0.40), and no significant correlation with the dialysis duration (*r* = 0.006, *r* = 0.944).

Using the T test, we report that in the studied group we could not find any differences, concerning the TSH serum concentration between the group with nodular goiter and the group without (*p* = 0.166). In respect to FT3 and FT4 serum concentrations, we did not find any differences with FT3 concentration (*p* = 0.075), but significant differences were found for FT4 concentrations (*p* = 0.016).

Thyroid volume negatively correlates with Kt/v ratio (*r* = −0.222, 95%CI = −0.388; −0.044, *p* = 0.014) (Figure 2) and positively with the body mass index (BMI) (*r* = 0.295, 95%CI = 0.062; 0.470; *p* = 0.001) (Figure 3). We could not find any significant interrelation between thyroid volume and age (*r* = −0.136; *p* = 0.135) or dialysis duration (*r* = −0.032; *p* = 0.726).

The presence of nodular goiter does not present significant difference when comparing to thyroid volume (*p* = 0.598). We did not find any differences in the nodular goiter group compared to the group without goiter in rapport with PTH serum concentration, serum creatinine, serum urea, Kt/v ratio, age, and BMI or dialysis duration.

The prevalence of thyroid disease in patients with ESRD on hemodialysis in the studied group, taking in account both nodular goiter and autoimmune thyroid disease is 61.8%. 

Using cross-tabulation and dividing dialysis duration into three groups (Group 1: 0–5 years of hemodialysis, Group 2: 5–10 years and Group 3: > 10 years of hemodialysis), the Chi-square test showed no association between the prevalence of thyroid disease and total dialysis duration (*p* = 0.552). The Chi-square test showed no association between the prevalence of autoimmune thyroid disease and total dialysis duration (*p* = 0.995) or between prevalence of nodular goiter and dialysis duration (*p* = 0.476).

We have found that the risk of developing thyroid disease in patients with end stage renal disease on hemodialysis is 3.4 higher in female gender (χ^2^ = 9.562; *p* = 0.002; OR = 3.41; 95%CI 1.541; 7.572); and the risk of developing nodular goiter is 2.5 times higher in female gender (χ^2^ = 6.233; *p* = 0.013; OR = 2.52; 95%CI 1.212; 5.280).

To assess the independent factors that predict thyroid disease for patients in the dialysis program we employed a backward multivariate logistic regression model. Akaike information criteria (AIC) were used to determine the best model, and the odds ratio and 95% confidence interval were calculated. Our regression equation proved to be a good fit for the model, explaining 15.5% of thyroid disease events (*R*^2^ = 0.155) (Table 4).

Thyroid disease risk is increased by 3.4-fold for the female gender and the increase of BMI with one unit raises the risk of developing thyroid disease with 1.083 times (*p* = 0.018).

## 4. Discussions

In this cross-sectional study, our main goal was to evaluate the prevalence of thyroid disease, both nodular goiter and autoimmune thyroid disease among patients with chronic kidney disease on hemodialysis. The study included all patients with ESRD on RRT from the clinic, with a slightly increased number of male subjects. 

Most of the patients were euthyroid (72.4%), however, hypothyroidism is quite frequent among these patients (24.4%). We have to keep in mind that in approximately 20% of uremic patients can present a reduction of FT3 concentrations, associated with systemic acidosis, malnutrition, inter-current processes, time on dialysis and inflammatory markers [12,42,43,44], therefore patients on RRT the diagnosis of hypothyroidism should be made on repeated documented TSH elevation and not only based on FT3 or FT4 reduction [12]. 

Various studies have demonstrated the relation between chronic kidney disease and thyroid disorders, CKD affecting the metabolism of thyroid hormones and thus influencing thyroid morphology [45]. Literature studies have showed that hemodialysis can cause thyroid abnormalities in both function and also morphology in patients with end stage renal disease [22,43,46]. 

In respect to the laboratory results, we have found that a high number of patients (46.34%) presented low levels of T3. Low T3 syndrome or “euthyroid sick syndrome” is a consequence of chronic non-thyroidal illness caused by uremia and protein malnutrition [47]. Subsequently FT3 and/or FT4 serum concentrations are prone to deviations based on non-thyroidal illness and may not accurately reflect thyroid functional status [1].

In our study, the total prevalence of thyroid disease, including both nodular goiter and autoimmune thyroid disease was 61.8%. A total of 57 patients (46.34%) presented low T3 syndrome, but only 40 patients presented thyroid disease. The total risk of developing thyroid disease is 3.41 times higher for women with ESRD on hemodialysis than for men. Our findings are sustained by similar results reported in the literature [8,19,30,33,45,47,48,49] with a high prevalence of thyroid disease among patients on hemodialysis, especially in women.

We did not discuss in our study about the prevalence of secondary hyperparathyroidism, this topic and others on hyperparathyroidism was covered in previous articles [41,50]. However, we did not find any positive correlation between thyroid volume and serum PTH concentration.

In our study group, seven patients had total thyroidectomy after the evaluation and three presented thyroid carcinoma. Although studies are suggesting that among patients with ESRD, some factors such as secondary hyperparathyroidism could induce an immune dysfunction, and even if the mechanism is unclear, it can lead to an increase in the incidence of thyroid cancer [37,38,51]. 

Literature studies are suggesting that patients with chronic kidney disease on renal replacement therapy, especially patients with thyroid disorders on levothyroxine replacement therapy can have a higher rate of hypothyroidism caused by phosphate binders. Phosphate binders can impair levothyroxine absorption leading to hypothyroidism [52]. However, we did not study this aspect in our study; as only 14 patients were on levothyroxine replacement therapy.

There were limitations to the study. Firstly, this cross-sectional design study depended on a particular population and did not have a control group of normal thyroid tests for the general population. Secondly, the sample size is limited, further investigations on larger number of patients are needed. 

In conclusion, we have found that thyroid disorders have a higher incidence among patients with end stage renal disease in hemodialysis therapy. Female patients from this category have a 3.41 times higher chance of developing a thyroid disease. The risk of developing nodular goiter is two times higher in female patients. Interestingly, we have found that an increase of BMI with more than one unit increases the risk of developing thyroid disease with 1.083 times in patients with ESRD.

## 5. Conclusions

To conclude, this cross-sectional study aimed to quantify the prevalence of thyroid disease in end-stage kidney disease patients on hemodialysis therapy, especially the presence of nodular goiter. 

Thyroid morphological disorders were present in high percentages and thyroid goiter was the prevalent thyroid disorder among these patients. 

Chronic kidney disease and hemodialysis therapy have an impact on thyroid echo texture and morphology of thyroid gland.

We have found that thyroid disease has a prevalence of 61.3% among patients with ESRD on hemodialysis with a higher chance of developing thyroid disease for female gender, and the increase of body mass index raises the risk of developing thyroid disease by 1.083 times. 

Clinical surveillance, dietary counseling and routine screening for thyroid disorders are important and can improve quality of life and life expectancy in these patients.

## Figures and Tables

**Figure 1 diagnostics-10-00245-f001:**
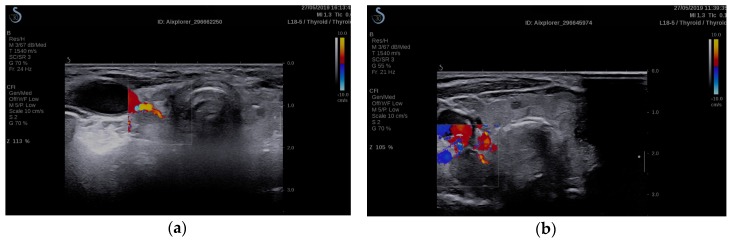
(**a**) 2B-Mode ultrasound evaluation of nodular goiter, visualizing parathyroid hyperplasia and thyroid nodule; (**b**) 2B-Mode ultrasound evaluation of autoimmune thyroiditis, visualizing parathyroid hyperplasia.

**Figure 2 diagnostics-10-00245-f002:**
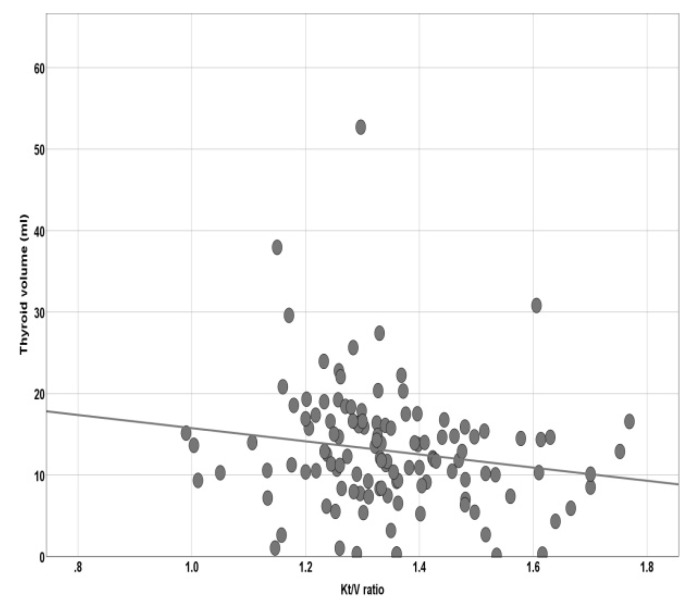
Pearson’s correlation between thyroid volume and Kt/v ratio.

**Figure 3 diagnostics-10-00245-f003:**
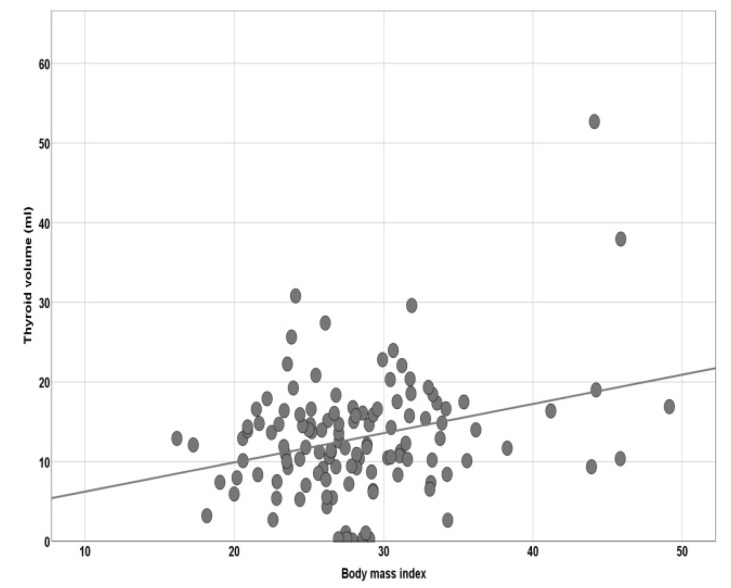
Pearson’s correlation of thyroid volume and BMI.

**Table 1 diagnostics-10-00245-t001:** Characteristic parameters of the study group.

Characteristics	Study Group
Gender (F/M)	53/70
Age (years)	62.2 ± 11.01
BMI (kg/m^2^)	28.3 ± 5.95
Hemodialysis duration (years)	5.6 ± 4.89
Parathormone (pg/mL)	609.4 ± 605.5
Total serum calcium (mg/dL)	8.6 ± 0.65
Serum creatinine (mg/dL)	8.9 ± 2.32
Serum urea (mg/dL)	122.2 ± 29.66
Kt/v ratio	1.3 ± 0.153
Glomerular filtration rate (mL/min/1.73 m)	5.5 ± 1.64
TSH (µU/mL)	3.36 ± 2.31
FT4 (pmol/L)	15.52 ± 2.92
FT3 (pmol/L)	3.2 ± 2.961
TG Ab (IU/mL)	16.0 ± 70.44
TPO Ab (IU/mL)	98.6 ± 295.8
Thyroid volume (mL)	13.3 ± 7.18

BMI—body mass index; TSH—thyroid stimulating hormone; TG Ab—thyroglobulin antibody; TPO Ab—peroxidase antibody.

**Table 2 diagnostics-10-00245-t002:** Frequency of thyroid disorders among the group.

Thyroid Disorders	Number of Patients	Percent Occurrence (%)
Euthyroid	89	72.4
Subclinical Hyperthyroidism	4	3.3
Hypothyroid	30	24.4

**Table 3 diagnostics-10-00245-t003:** Frequency of thyroid morphology disorders among the group.

Morphology Disorders	Number of Patients	Percent Occurrence (%)
Without Goiter	60	48.7
Nodular goiter	63	51.2
Autoimmune thyroiditis	22	17.8
Papillary thyroid carcinoma	3	2.4

**Table 4 diagnostics-10-00245-t004:** Assessment of independent factors that predict thyroid disease for patients in dialysis.

Variable	B	S.E.	*p*	OR	95% OR	
Female gender	1.276	0.416	0.002	3.581	1.585	8.092
BMI	0.90	0.038	0.015	1.083	1.005	1.168

BMI—body mass index; B—unstandardized beta; S.E.—standard error; p—p value; OR—odd ratio; 95% OR—95% odd ratio.

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
