# Peer review of "Thyroid Pathology in End-Stage Renal Disease Patients on Hemodialysis"

_diagnostics, 2020, doi:10.3390/diagnostics10040245_

Round 1

Reviewer 1 Report

In the manuscript "Thyroid pathology in ESRD patients on renal replacement therapy" Cotoi L. and colleagues performed a cross-sectional study in order to  investigate the prevalence of thyroid diseases in patients with chronic kidney disease (mostly at end stage), subjected to hemodialysis. Even though with several limitations, the work has certainly a clinical significance. However, some major and minor criticisms are present, as follows:

- The authors use  ESRD and CDK acronyms indiscriminately; it would be more appropriate to choose a definition to facilitate the reader;

- The title seems uninformative; please check whether "on renal replacement therapy" has the correct meaning;

- In general, the text is very confusing and the description does not follow a logical thread; Introduction: lines 58-64 (too short) are poorly connected to the subsequent part; paragraph 65-101: it is not clear whether the authors describe the effects of thyoid disorders on renal functionality in general, or in CDK patients; but at the end they deal with CDK and disturbances in hypothalamus -pituitary-thyroid axis; the authors should better organize this topic; the same is true for the paragraph (lines 102- 126 ), where the authors put a list of literature data without contextualizing with the topic of the manuscript;

- Tables 1-2-3 should be cited within the text;

- Figures lack of any legends;

- Lines 223-226: there is no correspondence with the text and the figures;

- Figures 1 and 2 are apparently unnecessary. What information do they give to support  authors' hypothesis?;

- The description of results from Table 4 seems a little inconsistent; moreover it's not clear what IMC refers to;

- Conclusions should be better described in order to highlight the significance and innovation of the work.

Author Response

Review 1

Honorable reviewer,

Thank you for your review suggestions, we have found them very helpful.

  1. The authors use ESRD and CDK acronyms indiscriminately; it would be more appropriate to choose a definition to facilitate the reader;

We have corrected and offered a clear definition, as we were referring about patients with end stage renal disease on hemodialysis.

“Chronic kidney disease (CKD) represents a worldwide public health problem and it is defined by the National Kidney Foundation (NKF) Kidney Disease Outcome Quality Initiative (K/DOQI) as kidney damage for more than three months with structural or functional abnormalities, with or without decreased glomerular filtration rate (GFR), manifested by pathological abnormalities or markers of kidney damage, or by GFR<60 ml/min/1.73 m2 for more than three months. CKD is classified into five stages upon GFR value [2]. Kidney Disease Improving Global Outcomes (KDIGO) guidelines characterize CKD by using markers of kidney damage (proteinuria and glomerular filtration rate). Chronic kidney disease is defined by the presence of both glomerular filtration rate less than 60mL/min and albumin greater than 30mg per gram of creatinine,  along with abnormalities of kidney structure or function for more than three months [3].  End-stage renal disease (ESRD) is defined as a GFR less than 15mL/min or CKD stage 5 [3,4].”

  1. The title seems uninformative; please check whether "on renal replacement therapy" has the correct meaning;

We have corrected the title, to be as clear as possible and have added missing information into the text.

“There is a wide array of methods to support the renal function, by delivering ongoing organ support. Renal replacement therapy (RRT) is defined by using extracorporeal methods (hemodialysis) or paracorporeal methods (peritoneal dialysis)[2,5].”

  1. In general, the text is very confusing and the description does not follow a logical thread; Introduction: lines 58-64 (too short) are poorly connected to the subsequent part; paragraph 65-101: it is not clear whether the authors describe the effects of thyoid disorders on renal functionality in general, or in CDK patients; but at the end they deal with CDK and disturbances in hypothalamus -pituitary-thyroid axis; the authors should better organize this topic; the same is true for the paragraph (lines 102- 126 ), where the authors put a list of literature data without contextualizing with the topic of the manuscript;

We have reorganized the text, to be as clear as possible, and focused our description on the effect of thyroid disease in end-stage renal disease and also the prevalence and thyroid hormone dysfunctions reported in the literature .

“Chronic kidney disease exhibits multiple endocrine and metabolic effects. One of the most common is on bone metabolism, causing chronic kidney disease - mineral bone disorder (CKD-MBD) and secondary hyperparathyroidism [6]. Another important endocrine pathology present in patients with chronic kidney disease is hypothyroidism and nodular goiter. There are important studies in the literature that support this fact [7–11]. The presence of nodular goiter in patients with chronic kidney disease brings an important burden on the clinician when facing a patient with secondary hyperparathyroidism. Evaluation of the percentage of thyroid disease, especially nodular thyroid goiter among patients with CKD could contribute to the correct diagnosis and treatment of these patients.

The interaction between the thyroid and the kidney in each other's functions is known for many years[12]. Thyroid hormones (TH) play an important role in kidney physiology. Their actions on growth, development and maintenance of kidney homeostasis are known and have been discussed in literature studies [13]. Thyroid dysfunctions, both hypothyroidism, and hyperthyroidism have an important effect on kidney function, as well as cardiovascular alterations [13].

Chronic kidney disease causes disturbances in the hypothalamus-pituitary–thyroid axis and TH peripheral metabolism [12,14–16]. Studies have shown that uremia plays an important role in patients with ESRD, having a direct relationship with the size of thyroid and the prevalence of thyroid goiter [7,12,17]. Also, thyroid nodules and thyroid cancer have a higher prevalence in uremic patients, than in the general population, especially in women [13,18].

Peripheral thyroid hormones (T3 and T4) can present low concentrations in serum, because of the hormonal removal during dialysis, decreased T3-binding capacity, altered hormonal catabolism, increased iodine store in thyroid gland, serum thyroid autoantibodies, and decreased peripheral conversion [7,12,13,19]. Low T3 syndrome has been observed in patients with ESRD [7,12,16,20], caused on one hand by the diminished peripheral conversion of T3 from T4 and on the other hand by the chronic metabolic acidosis [21,22].

End-stage renal disease can overlap the symptoms of hypothyroidism, such as fatigue, lethargy and cognitive dysfunction, as the prevalence data previously known are on relatively small cohorts and there are little data about the severity of thyroid abnormalities in these patients. A relatively high prevalence of thyroid goiter and an increase in thyroid gland have been reported [31]. An increased prevalence of hypothyroidism and subclinical hypothyroidism has been suggested in patients with ESRD [7,31]. Thyroid hormone dysfunction has been reported in ESRD patients, including low peripheral thyroid hormones and thyroid stimulating hormone[7,32].

The general population prevalence of overt hypothyroidism accounts for 0.2% to 5.3% in Europe [33,34] and 0.3% to 3.7% [35,36] of the general population. In the CKD population, there is a  high prevalence of primary hypothyroidism, as it increases with the decline of glomerular filtration rate (GFR) [37], especially in the subclinical form of hypothyroidism. Literature studies have shown a prevalence of 7% of subclinical hypothyroidism in patients with estimated GFR < 90 ml/min/1.73 m2, and the prevalence increased to 17.9% in subjects with GFR < 60 ml/min per 1.73 m2 [38–40].

The prevalence of hypothyroidism among ESRD patients on hemodialysis (HD) in previous studies was variable. One study reported a prevalence of subclinical hypothyroidism in HD patients of  21.8% compared with 7.14% in the control group[38], another study reported a prevalence of hypothyroidism in ESRD patients of 2.6% compared to 1.1% in the control group[7]. A study conducted on Iranian patients reported a prevalence of nodular goiter of 27.9% in HD patients versus 3.5% in the control group and hypothyroidism in 18.6% in HD patients versus 8.1% in the control group[41].

The prevalence of hyperthyroidism among patients with CKD is similar to the prevalence in the general population, around 1% [37,42,43].

Thyroid cancer prevalence was reported higher among ESRD patients on hemodialysis, with a positive correlation between the occurrence of thyroid cancer and duration of dialysis, and a higher rate of multifocality and also in post-transplant patients [44,45].

This study was designed to evaluate the prevalence of thyroid morphological changes and hormonal dysfunction in hemodialysis patients and to investigate if there are any associations between dialysis duration, age, body mass index or biochemical serum changes and the prevalence of thyroid disease.”

4.Tables 1-2-3 should be cited within the text;

We have correctly cited the table into the text.

  1. Figures lack of any legends;

We have added explicit legends to our figures.

  1. Lines 223-226: there is no correspondence with the text and the figures;

We have corrected the lack of correspondence of the figures with the text.

“Thyroid volume negatively correlates with Kt/v ratio (r=-0.222, 95%CI=-0.388;-0.044, p=0.014)  (figure 2) and positively with the body mass index (BMI) (r=0.295, 95%CI=0.062; 0.470; p=0.001)(figure 3).  We could not find any significant interrelation between thyroid volume and age (r=-0.136; p=0.135) or dialysis duration (r=-0.032; p=0.726).”

  1. Figures 1 and 2 are apparently unnecessary. What information do they give to support  authors' hypothesis?;

For figure one and two (now just figure 1) we added a specific paragraph to explain our hypothesis. The pictures displays the presence of both parathyroid adenomas and thyroid disorder, thus, knowing that patients with ESRD on dialysis can have a higher prevalence of thyroid disorders, we can have less false positive hyperplastic parathyroid glands.

“Secondary hyperparathyroidism was found in 59 patients, with a total of 97 hyperplastic parathyroid glands were visible on ultrasound [47]. In this group, 30 patients also presented nodular goiter (figure 1) and 13 patients presented autoimmune thyroiditis. Thyroid nodules frequently pose a dilemma to the clinician, firstly in respect to the nature of the nodule and also when faced with patients with parathyroid hyperplasia. The prevalence of thyroid disease, especially nodular thyroid goiter among patients with ESRD could contribute to the correct diagnosis and treatment of these patients.”

  1. The description of results from Table 4 seems a little inconsistent; moreover it's not clear what IMC refers to;

We have corrected the IMC, which referred to the patient BMI. In table 4 we present the assessment of independent  factors that predict thyroid disease for patients on dialysis.

  1. Conclusions should be better described in order to highlight the significance and innovation of the work.

We did try to better describe the utility and significance of our work, emphasizing the high prevalence of thyroid disease, especially in female patients and also the rise of the risk with an increase of body mass index, highlighting the importance of clinical surveillance, routine screening and dietary counseling.

“To conclude, this cross-sectional study aimed to quantify the prevalence of thyroid disease in end-stage kidney disease patients on hemodialysis therapy, especially the presence of nodular goiter.

Thyroid morphological disorders were present in high percentages and thyroid goiter was the prevalent thyroid disorder among these patients.

Chronic kidney disease and hemodialysis therapy have an impact on thyroid echo texture and morphology of thyroid gland.

 We have found that thyroid disease has a prevalence of 61.3% among patients with ESRD on hemodialysis with a higher chance of developing thyroid disease for female gender, and the increase of body mass index raises the risk of developing thyroid disease by 1.083 times.

Clinical surveillance, dietary counseling and routine screening for thyroid disorders is important and can improve quality of life and life expectancy in these patients.”

We did make a lot of spelling and grammar mistakes, we have revised our manuscript for spelling and grammar errors.

We hope that the changes are according to your recommendations.

Thank you,

The authors.

Reviewer 2 Report

Dear Authors,

Please find listed below major concerns regarding this manuscript:

  1. Lack of novelty. You already mentioned (lines 287-288) that similar studies and similar results were already performed/presented (8 papers were cited). How does your data differ from what has already been published?
  2. Lack of important controls (control group of normal thyroids) and size of cohort (however, this subject was addressed in the discussion, lines 301-303).
  3. The results section is hard to follow – all the % and fold changes are not clear – data should be presented in tables etc.
  4. ESRD not ERSD? please check the manuscript and correct. End-stage renaldisease (ESRD).

Author Response

Review 2

Honorable reviewer,

Thank you for your review suggestions, we found them very helpful.

  1. Lack of novelty. You already mentioned (lines 287-288) that similar studies and similar results were already performed/presented (8 papers were cited). How does your data differ from what has already been published?

There are similar literature studies, however in our study we have found that not only thyroid disorders are more prevalent in women, who have a 3.41 x times higher risk for thyroid disease than men.  We have rephrased the mentioned paragraph, in order to better emphasize our results.

“In our study, the total prevalence of thyroid disease, including both nodular goiter and autoimmune thyroid disease was 61.8%. A total of 57 patients (46.34%) presented low T3 syndrome, but only 40 patients presented thyroid disease.The total risk of developing thyroid disease is 3.41 x times higher for women with ERSD on hemodialysis than for men.  Our findings are sustained by similar results reported in the literature [12,20,38,41,51,53–55] with a high prevalence of thyroid disease among patients on hemodialysis, especially in women.”

  1. Lack of important controls (control group of normal thyroids) and size of cohort (however, this subject was addressed in the discussion, lines 301-303).

The lack of controls represents a limitation to our study.

  1. The results section is hard to follow – all the % and fold changes are not clear – data should be presented in tables etc.

The percentual occurance of thyroid disorders and morphology disorders are presented in table 2, respectively in table 3.

We have restructured the results section, in order to present in clearer manner.

  1. ESRD not ERSD? please check the manuscript and correct. End-stage renal disease (ESRD).

There were typos for ESRD, we have checked the manuscripts and corrected them.

We did make a lot of spelling and grammar mistakes, we have revised our manuscript for spelling and grammar errors.

We hope that the changes are according to your recommendations.

Thank you,

The authors.

Round 2

Reviewer 1 Report

The authors have answered  sufficiently my questions. The manuscript has been improved and may be accepted for publication.

Reviewer 2 Report

Dear Authors, Thank you very much for the fast correction of the text and responses. The article in the present form is acceptable for publication. I found a few minor "typo" errors, which can easily be corrected.